# Sparse Feature Learning of Hyperspectral Imagery via Multiobjective-Based Extreme Learning Machine

**DOI:** 10.3390/s20051262

**Published:** 2020-02-26

**Authors:** Xiaoping Fang, Yaoming Cai, Zhihua Cai, Xinwei Jiang, Zhikun Chen

**Affiliations:** 1The Department of Computer Science, China University of Geosciences, Wuhan 430074, China; fangxiao_ping@126.com (X.F.); ysjxw@hotmail.com (X.J.); 2The Beibu Gulf Big Data Resources Utilization Laboratory, Beibu Gulf University, Qinzhou 535011, China; chzhikun@163.com; 3The Guangxi Key Laboratory of Marine Disaster in the Beibu Gulf, Beibu Gulf University, Qinzhou 535011, China

**Keywords:** extreme learning machine autoencoder, autoencoder, evolutionary multiobjective optimization, sparse feature learning, hyperspectral imagery

## Abstract

Hyperspectral image (HSI) consists of hundreds of narrow spectral band components with rich spectral and spatial information. Extreme Learning Machine (ELM) has been widely used for HSI analysis. However, the classical ELM is difficult to use for sparse feature leaning due to its randomly generated hidden layer. In this paper, we propose a novel unsupervised sparse feature learning approach, called Evolutionary Multiobjective-based ELM (EMO-ELM), and apply it to HSI feature extraction. Specifically, we represent the task of constructing the ELM Autoencoder (ELM-AE) as a multiobjective optimization problem that takes the sparsity of hidden layer outputs and the reconstruction error as two conflicting objectives. Then, we adopt an Evolutionary Multiobjective Optimization (EMO) method to solve the two objectives, simultaneously. To find the best solution from the Pareto solution set and construct the best trade-off feature extractor, a curvature-based method is proposed to focus on the knee area of the Pareto solutions. Benefited from the EMO, the proposed EMO-ELM is less prone to fall into a local minimum and has fewer trainable parameters than gradient-based AEs. Experiments on two real HSIs demonstrate that the features learned by EMO-ELM not only preserve better sparsity but also achieve superior separability than many existing feature learning methods.

## 1. Introduction

Hyperspectral Imagery (HSI), which is obtained by remote sensing systems, contains high-resolution spectral information over a wide range of the electromagnetic spectrum with hundreds of observed spectral bands [1]. The detailed spectral and spatial information provides the power of accurately differentiating or recognizing materials of interest [2]. In recent years, HSI has been applied in a wide variety of applications, including agriculture, surveillance, astronomy, and biomedical imaging, among others [3]. However, a great number of redundancies between spectral bands bring heavy computation burdens in HSI data analysis [4]. Furthermore, high dimensionality even rises “Hughes” problem, namely the so-called curse of dimensionality [5,6].

As an effective solution, feature learning overcomes these issues and guarantees good classification accuracy [7,8,9,10,11,12,13]. The conventional feature learning methods, such as Principal Component Analysis (PCA) [9] and its variants [14,15,16] are widely applied in HSI. However, PCA is a linear combination of all the original data, thus it is often difficult to interpret the results. To improve PCA’s performance, Sparse PCA (SPCA) [11] was proposed using the lasso ( elastic net) to produce modified principal components with sparse loadings. Besides that, Autoencoder (AE) [17] is a usual recent feature learning method. AE is a well-known unsupervised neural network which is composed of an encoder and a decoder. AE aims at learning an identity transformation for raw inputs [18]. The learned features are generally represented as outputs of the encoder (the hidden layer). In many applications, labels of hyperspectral imagery are expensive and difficult to obtain, thus AE can work as an effective unsupervised feature learning approach and be widely applied in the hyperspectral image [19,20,21,22]. However, AE is faced with its instinct limitations that need to be solved. On the one hand, AE has to simultaneously update the parameters for encoder and decoder, that means the high cost of parametric learning. On the other hand, the vanishing gradient problem exists in the training of AE that bases on the backpropagation algorithm [23]. To solve it, sparse AE(SAE) was proposed by imposing sparse constraints on the encoder to guarantee the sparsity of coding results [24]. In this case, new hyperparameters could be introduced so that it is difficult to determine the best hyperparameters. However, parameters of AE and SAE need to be adjusted frequently, resulting in high cost of learning time.

Extreme Learning Machine( ELM) was originally proposed for generalized single-hidden layer feedforward neural networks [25,26,27]. Due to its fast learning speed, good generalization ability, and ease of implementation, ELM has become a popular research topic and been widely used for supervised learning of HSI [28,29,30,31,32]. Although ELM is successfully used for supervised learning in hyperspectral image classification, it is also difficult to avoid a large number of labeled samples. However, the number of labeled samples is so small because of expensively and difficultly labeling for samples in the hyperspectral imagery. Thus, an unsupervised feature learning approach is strongly desired to solve it. But unsupervised feature learning based on ELM has not drawn much attention, the reason for which is mainly that ELM is limited by its hidden layer parameters that are generated randomly. Actually, in terms of AE, ELM can be viewed as a Nonlinear Random Projection (NRP) combined with a Linear Regression (LR). The non-optimal NRP weights could result in its outputs including noise and redundancies. Compared with AE, ELM can learn quickly without adjusting parameters. However, ELM is difficult for unsupervised learning. Based on the above consideration, ELM-AE [33] is proposed by taking into account the merits of AE and ELM, and applied to unsupervised feature learning in hyperspectral imagery. Compared with AE, ELM-AE only deals with the parameters of an encoder, instead of decoder, which means the complexity of the problem is decreased. Meanwhile, it preserves the strong capability to extract features. Besides, it can handle the unsupervised feature learning for hyperspectral imagery that is what the basic ELM can’t have. According to the advantages mentioned above, ELM-AE is often used for unsupervised learning and constructing multilayer ELM [34,35,36]. In [37], ELM-AE has been proven effective in dimension reduction. To increase the sparsity of parameters, Sparse Random Projection (SRP) [38,39] was introduced into ELM-AE and provided better generalization ability. ELM-AE and sparse ELM-AE(SELM-AE) essentially are a linear transformation of initial features using LR’s weights as a transform matrix. However, according to the above-mentioned methods, the parameters of the hidden layer are found out in the matrix transformation way, which is not suitable for the fact that a large number of nonlinear data exist in the hyperspectral image. In the global data domain, Evolutionary Multiobjective Optimization(EMO) can obtain the nonlinear solution in the form of parallel. Evolutionary Multiobjective-based ELM(EMO-ELM) is proposed to solve the nonlinear problem of ELM-AE and SELM-AE.

The motivation of this paper is to propose a novel sparse unsupervised feature learning approach, and then deal with nonlinear data evolutionarily. Based on the consideration above, our paper proposes a novel AE framework for sparse feature learning by treating NRP and LR in ELM-AE as encoder and decoder. Different from ELM-AE, the proposed method provides the optimal weights and bias by EMO. Similar to the conventional AE, we use the outputs of encoder to represent the learned feature. The decoder’s weights are directly calculated through the ELM theories, thus we only need to update the parameters in the NRP. Our goal is to reduce the parameters involved in AE and improve ELM-AE. Due to that the gradient-based methods are disabled in our method, we design a multiobjective model composed of two conflicting objectives, including the sparsity of learned features and reconstruction error. In order to solve this two-objective model, EMO is effectively employed to handle this type of optimization problem. Comparing with AE and SAE, the number of parameters have been reduced to half; comparing with ELM-AE and SELM-AE, this approach learns features by a nonlinear manner, moreover, the using of optimization method avoids the influence of NRP. Through constructing two conflicting objectives composed of the sparsity and reconstruction error, and extracting features from the selected solutions based on optimal Pareto Front (PF) obtained by EMO, EMO-ELM improves the classification accuracy while ensuring the sparsity. Experimental results verify that the purposed method outperforms NRP, SPCA, AE, SAE, ELM-AE and SELM-AE in terms of the sparsity and reconstruction error.

The main contributions of this paper can be summarized as follows: (1) A novel unsupervised feature learning approach is proposed; (2) We combine the advantages of basic ELM and conventional AE; (3) A hybrid feature learning framework is presented which uses EMO for training.

The remainder of this paper is organized as follows. Section 2 reviews the related work, including ELM-AE and AE. Section 3 describes the proposed EMO-ELM method in detail. The experimental results and discussions are shown in Section 4. Finally, the conclusions are represented in Section 5.

## 2. Related Work

### 2.1. ELM-AE

ELM-AE can be divided into NRP and LR, and its structure is shown in Figure 1a. Let us consider *N* distinct original samples $X={\left\{x_{i}\right\}}_{i=1}^{N}$, where in $x_{i}\in R^{n}$ (n-dimensional feature space). For example, ELM-AE is used to extract features of SalinasA data set for comparison in this paper, then N=86×83 and n=204. Supposing the dimension of the extracted feature $\tilde{X}$ is 100, $\tilde{X}$ is the preprocessed data of N=86×83 samples that have the same number of samples as the original data *X*. The input and output layers have the same number of neurons and the input and output vectors have the same dimension, which is *n*. The input layer is equivalent to the encoding part of the automatic encoder, while the output layer to the decoding part. Further supposing that the number of hidden neurons *L* is less than the number of hidden neurons *n*, the weight matrix is orthogonal, we see that the first part of ELM-AE aims at mapping *X* into an *L*-dimensional space by the following equation:(1)$$\left\{\begin{array}{c} \begin{array}{c} h_{i}=g\left(W^{T}x_{i}+b\right),i=1,\cdots ,N\\ W^{T}W=I\\ b^{T}b=1 \end{array} \end{array}\right.$$
where $h_{i}={\left[h_{i1},h_{i2},\cdots ,h_{iL}\right]}^{T}\in R^{L}$, $x_{i}={\left[x_{i1},x_{i2},\cdots ,x_{in}\right]}^{T}\in R^{n}$, $W=\left[w_{1},w_{2},\cdots ,w_{L}\right]\in R^{n\times L}$ and $b={\left[b_{1},b_{2},\cdots ,b_{L}\right]}^{T}\in R^{L}$ denotes the output vector of a hidden layer for the i-th input sample, the vector of the i-th input sample, NRP’s orthogonal weights matrix connecting the neurons of the input layer with the neurons of the hidden layer, and the orthogonal bias vector in the hidden layer respectively. The output weights β of ELM-AE are responsible for the transformation from the feature space to input data and satisfy:(2)$$\underset{\beta }{\mathrm{Minimize}}||H\beta -{X||}^{2}$$

According to ELM-AE’s theories, the output weights β of ELM-AE are:(3)$$\beta =W^{T}VV^{T}$$
where V is the eigenvectors of covariance matrix $X^{T}X$.

The new feature can be achieved in ELM-AE by projecting the data X along the decoder stage weight β as:(4)$$\tilde{X}=X\beta^{T}$$$\tilde{X}$ is the new feature and can be applied to hyperspectral image classification.

When the number of input neurons in ELM-AE is smaller than the number of hidden neurons (n<L), the weight matrix is sparse. According to the ELM theories, W and b can be randomly generated. $g\left(\cdot \right)$ is any nonlinear activation function, such as Sigmoid function. To further increase the model’s sparsity, W and b, which are orthogonal, can be generated using the following equation [37]:(5)$$\begin{array}{c} w_{ij}=b_{j}=1/\sqrt{L}\times \left\{\begin{array}{cc} +\sqrt{3} & p=1/6\\ 0 & p=2/3\\ -\sqrt{3} & p=1/6 \end{array}\right. \end{array}$$
where $w_{ij}$ indicates the weight connecting the *i*-th neuron in the input layer with the *j*-th neuron in the hidden layer, $b_{j}$ is the bias of the *j*-th neuron in the hidden layer, and *p* represents the ratio of elements in the sparse random matrix W. $w_{ij}$ and $b_{j}$ are generated randomly according to the *p*-value, similar to the wheel algorithm. It is assumed that there is a random probability *r* before every matrix element is generated, and the value range of *r* is between 0 and 1. The elements in the matrix change according to the random generation probability *r*. When the probability *r* is less than 1/3, the values of $w_{ij}$ and $b_{j}$ are $\sqrt{\frac{3}{L}}$; when the random probability *r* is more than 1/3 and less than 5/6, the values of $w_{ij}$ and $b_{j}$ are 0; when the random probability *r* is greater than 5/6 and less than 1, the values of $w_{ij}$ and $b_{j}$ are $-\sqrt{\frac{3}{L}}$.

The output weights β of SELM-AE are calculated as:(6)$$\underset{\beta }{\mathrm{Minimize}}||H\beta -{X||}^{2}$$

According to ELM-AE’s theories, the output weight β of SELM-AE are:(7)$$\beta =W^{T}VV^{T}$$
where V is the eigenvectors of covariance matrix $X^{T}X$.

Dimension reduction is achieved in SELM-AE by projecting the data X along the decoder stage weights β as:(8)$$\tilde{X}=X\beta^{T}$$

The second part of ELM-AE is a well-known ridge regression or regularized least squares, which aims to solve the output weights matrix β of which $\beta_{ji}$ indicates the weight connecting the *j*-th neuron in the hidden layer with the *i*-th neuron in the output layer by minimizing the following learning problem:(9)$$\underset{\beta }{\mathrm{Minimize}}:{∥\beta ∥}_{t}^{\sigma_{1}}+\lambda {∥H\beta -X∥}_{q}^{\sigma_{2}}$$

The first term in the objective function is the output weights matrix, and the second term represents the reconstruction errors. λ is a regularization term that controls the complexity of the learned model. Here, according to the ELM theorem [37,40], $\sigma_{1}>0,\sigma_{2}>0$, *t*, and *q* can be any matrix norm (e.g., $t,q=0,\frac{1}{2},1,2,\cdots ,+\infty$). To obtain the analytical solution, $\sigma_{1},\sigma_{2},t$ and *q* are often set to 2, which is also known as a ridge regression problem. However, t,q=0 (i.e., L0-norm) are possible. Under the circumstances, ${\left\|\beta \right\|}_{0}$ denotes the number of non-zero weights in the hidden layer, and ${\left\|H\beta -X\right\|}_{0}$ indicates that the amount of non-zero reconstruction errors for the original data X.

When the penalty coefficient λ is infinitely large in Equation (Equation 9), β can be calculated as
(10)$$\beta =H^{†}X$$
where $H^{†}$ indicates the Moore–Penrose generalized inverse of H. According to the ELM learning theory, ELM with a minimum norm of output weights has better generalization performance and a more robust solution [41].

Different output weights β can be obtained based on the concerns on the efficiency in different size of training data sets. When the number of training samples is not huge, the output weight β can be expressed as
(11)$$\beta =H^{T}{\left(HH^{T}+\frac{I}{\lambda }\right)}^{-1}T$$

The output function of ELM classifier is
(12)$$f\left(x\right)=h\left(x\right)\beta =h\left(x\right)H^{T}{\left(HH^{T}+\frac{I}{\lambda }\right)}^{-1}T$$

When the number of training samples is very large, the output weight β can be represented as
(13)$$\beta ={\left(H^{T}H+\frac{I}{\lambda }\right)}^{-1}H^{T}X$$

And the output function of ELM classifier is
(14)$$f\left(x\right)=h\left(x\right)\beta =h\left(x\right){\left(H^{T}H+\frac{I}{\lambda }\right)}^{-1}H^{T}T$$

Subsequently, the new feature $\tilde{X}$ can be represented as:(15)$$\tilde{X}=X\beta^{T}$$

### 2.2. AE

AE has a symmetrical encoder-to-decoder structure as seen in Figure 1b. For the sake of simplicity, we represent AE’s cost function as
(16)$$J\left(\Theta_{e},\Theta_{d}\right)=L\left(X-F_{d}\left(F_{e}\left(X;\Theta_{e}\right);\Theta_{d}\right)\right)+\lambda R\left(F_{e}\right)$$where L is the loss term such as Mean Squared Error (MSE); R is the regularization term with the regularization coefficient λ, in SAE, R can be written as ${∥\Theta_{e}∥}_{p}$ or ${∥F_{e}\left(X;\Theta_{e}\right)∥}_{p}$; $F_{e}$ and $F_{d}$ denote encoder and decoder’s transformation with hyperparameters $\Theta_{e}$ and $\Theta_{d}$, respectively. The training of AE can be finished using the gradient backpropagation algorithms.
(17)$$\Theta =\Theta -\mu \frac{\partial J}{\partial \Theta },\;\Theta =\left[\Theta_{e},\Theta_{d}\right]$$

Here μ denotes the learning rate, and $\frac{\partial J}{\partial \Theta }$ indicates the gradient of Θ. Unlike ELM-AE, AE represents the learned features using the outputs of an encoder and it can be written as
(18)$$\tilde{X}=F_{e}\left(X;\Theta_{e}\right)$$

Because $F_{e}$ is generally nonlinear, AE can process complicated data. For example, AE has been successfully used for HSI feature learning and classification in [24].

## 3. EMO-ELM

The structure of the proposed approach is given in Figure 2. We still keep ELM-AE’s structure and parameter settings but use parametric learning steps by encoder-to-decoder like AE. In our approach, the gradient-based tuning method is difficult to work, therefore a multiobjective-based method is taken into consideration. In this section, we first introduce the multiobjective construction; next, we give the Non-dominated Sorting Genetic Algorithm II(NSGA-II)-based learning steps; then we describe the solution selection procedure; finally, we use the proposed approach to represent the sparse feature.

### 3.1. Constructing a Multiobjective Model

Our multiobjective model contains two conflicting objective functions with respect to the decision variable vector α, where $\alpha ={\left[w_{11},\cdots ,w_{1L},\cdots ,w_{n1},\cdots ,w_{nL},b_{1},\cdots ,b_{L}\right]}^{T}$, and we indicate them as $f_{1}$ and $f_{2}$, respectively. The detailed definitions of $f_{1}$ and $f_{2}$ are given below.

Firstly, we define $f_{1}$ as the sparseness of the encoded results. Suppose the activation function of the hidden neurons to be the Sigmoid function, then the activation values in the hidden layer will be limited in $\left(0,1\right)$. Informally, we think of a hidden neuron as being “active” (or as “firing”) if its output value is close to 1, or as being “inactive” if its output value is close to 0. The lower the probability of the activation function among the hidden layer, the sparser the encoded results, and the smaller $f_{1}$ is. Our goal is to constrain the hidden neurons to be inactive most of the time. We first define the average activation of the *j*-th hidden neuron as
(19)$${\hat{\rho }}_{j}=\frac{1}{N}\sum_{i=1}^{N}g_{ij}\left(x_{i}\right)=\frac{1}{N}\sum_{i=1}^{N}h_{ij}$$

Further, let ρ be the desired activation, our goal is forcing ${\hat{\rho }}_{j}$ to close to ρ. To measure the difference between ρ and ${\hat{\rho }}_{j}$, the Kullback-Leibler(KL)-divergence is takeninto consideration, which can be expressed as follows
(20)$$\mathrm{KL}\left(\rho \|{\hat{\rho }}_{j}\right)=\rho \log\frac{\rho }{{\hat{\rho }}_{j}}+\left(1-\rho \right)\log\frac{1-\rho }{1-{\hat{\rho }}_{j}}$$

Considering all the hidden neurons, $f_{1}$ is represented as the sum of the KL-divergence of all hidden neurons.
(21)$$f_{1}=\sum_{j=1}^{L}\mathrm{KL}\left(\rho \|{\hat{\rho }}_{j}\right)$$

Secondly, we define $f_{2}$ as the Root Mean Squared Error (RMSE) of the raw input and the decoded results. However, the decoder determines parameters mathematically, we adopt K-fold crossover validation to calculate RMSE to further avoid overfitting. Thus, $f_{2}$ can be denoted as
(22)$$f_{2}=\frac{1}{K}\sum_{k=1}^{K}\left(\sqrt{\frac{\sum_{i=1}^{n_{k}}{∥x_{i}-y\left(x_{i}\right)∥}_{2}^{2}}{n_{k}\times n}}\right)$$
where *K* indicates a *K*-fold crossover validation is used, $n_{k}$ is the number of validation samples in *k*-th fold subjected to $\sum_{k=1}^{K}n_{k}=N$, and $y\left(x_{i}\right)$ is the decoded results for $x_{i}$. Ultimately, we aim to simultaneously minimize the two functions. For clarity, the overall multiobjective model is expressed as
(23)$$\underset{\alpha }{\arg\min}\;F\left(\alpha \right)=\underset{\alpha }{\arg\min}\;\left(f_{1},f_{2}\right)$$

### 3.2. Solving a Multiobjective Model

To effectively solve the optimization problem given in Equation (Equation 23), we employ EMO algorithms. Specifically, NSGA-II [42] is used in this paper due to its fast solvingspeed, excellent convergence, and popular applications. All kinds of evolutionary multiobjective algorithms such as Multi-Objective Evolutionary Algorithm based on Decomposition (MOEA/D) [43] and Multi-Objective Particle Swarm Optimization(MOPSO) [44], are practicable in our framework. The mainsteps using NSGA-II to solve our multiobjective model are given in Algorithm 1.
**Algorithm 1:** NSGA-II-based Solving   **Input:**
NP, gen, other evolutionary parameters   **Output:** Pareto optimal solution set1 Initialize population;2 **while**
*Termination criteria not met*
**do**3

Elitist selection technique;4Generic operations;5Objectives evaluation;6Fast nondominated sorting;7Crowding distance assignment;8 **end**

### 3.3. Selecting Solution

By evolving, we get a set of solutions, which is called Pareto optimal solutions. Generally, the corresponding objective values can be plotted as a curve named PF curve. Unlike single-objective optimization method, we have to choose from the Pareto optimal set, because these solutions are deemed to be equally important. In this paper, the following three solutions are considered as alternatives.

the solution getting a minimum value of $f_{1}$;the solution getting a minimum value of $f_{2}$;the solution locating at the knee area.

The first two solutions are easy to determine. However, focusing on the knee area is commonly difficult, because the PF could be not smooth and the objective functions involved in the model usually have greatly different magnitudes. To observe the PF, we find a fact that the knee usually occurs concurrently with a maximum curvature of PF curve. Thus, this problem is transformed to find out the maximum curvature of the PF curve. To do this, the following steps are carried out. Firstly, normalize the Pareto solution to overcome the different magnitudes. Next, smooth the PF by interpolating the PF using B-splines. And then evenly resample from the smooth spline. Finally, estimate the curvature according to the derivatives of the B-spline curve and select the solution which is closest to the maximum curvature.

### 3.4. Sparse Feature learning Using EMO-ELM

We denote the selected solution as αF. Subsequently, we use αF. to regenerate the parameters of the encoder, which is represented as WF and bF. Depending on this, the learned features can be obtained from the following processing.
(24)X∼=gXWF+BF
where X∼ is the learned features, and BF=[bFT,⋯,bFT]N×LT. As seen in Equation (Equation 24), this procedure is a nonlinear transformation. The complete pseudocode of using EMO-ELM to extract sparse features is given in Algorithm 2.
**Algorithm 2:** EMO-ELM for sparse feature learning   **Input**: X, ρ, *L*   **Output**: Learned feature1 Optimize Equation (Equation 23) according to Algorithm 1, and obtain the Pareto optimal solution set;2 Select αF from the obtained Pareto optimal solution set according to selection criteria;3 Regenerate WF and bF;4 Extract features according to Equation (Equation 24);5 Return extracted features X∼;

## 4. Experiments

### 4.1. Data Description and Experiment Design

#### 4.1.1. SalinasA Data Set

This image was collected by the 224-band Airborne Visible/Infrared Imaging Spectrometer(AVIRIS) sensor over SalinasValley, California, and was characterized by high spatial resolution (3.7-meter pixels). The area covered comprises 86×83 pixels and includes six classes, of which the class information is given in Table 1. The pseudo-color image and its ground truth are shown in Figure 3. This scene has reduced the number of bands to 204 by removing 20 bands covering the region of water absorption: 108–112, 154–167, 224.

#### 4.1.2. Kennedy Space Center (KSC) Data Set

This scene was acquired by NASA AVIRIS (Airborne Visible/Infrared Imaging Spectrometer) instrument over the Kennedy Space Center (KSC), Florida, on March 23, 1996. This data is with 512×614 pixels and has a spatial resolution of 18 m, which was shown in Figure 4. Regardless of discarded water absorption and low signal-to-noise ratio (SNR) bands, 176 spectral bands are used for classification. 13 different land cover classes available in the original dataset are displayed in Table 2.

### 4.2. Experiment Settings

Experiments will be organized into three parts for the convergence, sparsity, and separability. The first one aims at analyzing the convergence of EMO-ELM, and further illustrating the solution selection. In the second experiment, we quantitatively compare the sparsity of EMO-ELMs (EMO-ELM($f_{1}$), EMO-ELM($f_{2}$) and EMO-ELM(best) represent EMO-ELM with different strategies of solution selection respectively) with NRP (namely unoptimized ELM), SPCA [11], ELM-AE [37], SELM-AE [37], AE, and SAE. Moreover, Finally, we use the basic ELM with 500 hidden neurons as the classifier to estimate the classification capabilities of the learned features in terms of Overall Accuracy (OA), Average Accuracy (AA), and Kappa coefficient. In this part, all experiments are executed through three-fold cross-validation and repeated ten times and take the mean as the performance criteria. 10 percent of samples are used for training and the rest is for testing. For AE and SAE, we use the implementation available from the Keras website [45]. For EMO-ELM, the well-known NSGA-II (https://github.com/Project-Platypus/Platypus) is applied, in which we uniformly set NP=50 and gen=5000. Additionally, all experiments are carried out using Python 2.7 on an Intel i5 Core(TM) 3.2-GHz machine with 8 GB of RAM.

### 4.3. Convergence and Solution Selection

The optimizing of Multi-Objective Problems(MOP) requires the objectives that need to be solved are conflicted to each other. Thus, we can determine whether the algorithm is convergent by observing the PF. We consider algorithm is convergent if all these solutions are non-dominated. At this point, the PF looks like a ’smooth’ curve. In Figure 5, we illustrate the normalized PFs obtained by optimizing 10-hidden-neurons ELM with 5000 generations on SalinasA and KSC data sets. As we can see, after 5000-generation evolution, the final results are completely convergent. Furthermore, it was revealed that two objectives we constructed are conflicted. Observing the curvature curves, the values will be enlarged dramatically in the knee area, therefore we can accurately find out it.

### 4.4. Visual Investigation of Features Learned by Different Algorithms

Iris data set, which contains 150 samples with three classes, is suitable for visualization. Figure 6a–i show the distribution of the features of Iris data projected into two-dimensional space. Observed from Figure 6a, the original nonlinear random projection in ELM disorders the data distribution, which indicates that there are included noise in the original ELM. As seen in Figure 6b–i, the features of Iris are effectively distributed in the feature space after feature learning. By performing EMO-ELM, these noises have been dramatically reduced. From the cluster point of view, EMO-ELM decreased the within-class distance and increased the between-class distance. This phenomenon is especially evident in Figure 6i which uses the proposed knee-based solution selection strategy.

### 4.5. Measuring Sparsity of the Learned Features

In order to quantify the sparsity, we use the $L_{2}/L_{1}$ sparsity measure, which is applied in [37,46]. $L_{2}/L_{1}$ measure indicates sparsity at an abstract level. A higher $L_{2}/L_{1}$ measure demonstrates that there are few large feature values and more small feature values. On the contrary, there are more large feature values and a few small feature values. In other words, the higher the $L_{2}/L_{1}$ gets, the sparser the learned feature is. In Figure 7, we give the $L_{2}/L_{1}$ sparsity measure of different algorithms under the various number of features. It is certain that the features learned by EMO-ELMs are sparser than all the competitors. Comparing with NRP, EMO-ELM has significantly increased the sparsity.

### 4.6. Comparison of Classification Ability

For feature learning, an important criterion for evaluating the effectiveness of learned features is classification ability. In this experiment, we compare EMO-ELM’s performance with the above-mentioned six feature learning approaches under 10-dimensional features. The results of SalinasA and KSC are given in Table 3 and Table 4. The best results are bold. As seen from Table 3 and Table 4, the EMO-ELMs, especially EMO-ELM($f_{2}$), outperforms other methods, which means that the performance could be more excellent in optimization way. To visually present the differences between competitors, we also plot the statistical evaluations of OA, AA, and Kappa in Figure 8a–c for SalinasA and (d–f) for KSC. As shown in Figure 8, EMO-ELM($f_{2}$) obtains the best results in comparison with other strategies in terms of all aspects, which is because that EMO-ELM($f_{2}$) acquires the least reconstruction error. The detailed conclusions are given as follows:EMO-ELM($f_{1}$) v.s. EMO-ELM($f_{2}$) v.s EMO-ELM(best): Generally, EMO-ELM($f_{2}$) yields higher accuracy than other competitors in terms of mean performance for SalinasA and KSC. The reason is that because EMO-ELM($f_{2}$) guarantees the model to achieve the smallest reconstruction error, whereas EMO-ELM($f_{1}$), although, obtains the smallest sparsity, the feature reconstruction is limited. Reviewing Figure 7a,b, EMO-ELM($f_{2}$) also maintains good sparsity. Hence, the solution selection strategy based on $f_{2}$ can be considered best in our experiments. EMO-ELM(best) also plays a trade-off role between sparsity and reconstruction error, thus we view it as the second choice.NRP v.s. EMO-ELM: As shown in Figure 8a–f, the original nonlinear random projection (NRP) is effective in feature mapping, but EMO-ELM has shown that the NRP’s performance can be further improved after optimizing.SPCA v.s EMO-ELM: The features learned by SPCA maintain the remarkable sparsity, however, the classification ability is damaged. Furthermore, as a dimension-reduction method, SPCA cannot work when the learned dimension is larger than the original dimension. On the contrary, EMO-ELM outperforms SPCA in respects of many tested performances.ELM-AE and SELM-AE v.s EMO-ELM: As we known, ELM-AE and SELM-AE learn features linearly. In this experiment, SELM-AE performs better than ELM-AE due to the sparse matrix is used. Whereas, EMO-ELM learns features nonlinearly and has significantly enhanced the classification capacity of the learned features.AE and SAE v.s EMO-ELM: The learning procedure of EMO-ELM is similar to AE and SAE, but EMO-ELM becomes more competitive in both respects of classification ability and sparsity after the same times of updating. Especially, EMO-ELM optimizes only the hidden layer, whereas AE and SAE have to simultaneously optimize the hidden layer and the output layer.

### 4.7. Discussion

Based on the above experimental results, we provide a meaningful discussion on this article:The proposed approach can be regarded as a general framework that is composed of a nonlinear encoder and a linear decoder. For the optimization of this framework, we only need to focus on the encoder (or the hidden layer) since the decoder can be represented as a closed-form solution, which is very different from neural networks. Thus, compared with SAE and AE, the number of EMO-ELM’s parameters can be reduced to half.In addition to the objectives used in this paper, various objectives, such as classification error and matrix norm constraints are considered in this framework. More importantly, the optimizer is replaceable and flexible. Therefore, EMO-ELM can be used as an alternative to unsupervised feature learning.It is well known that the evolutionary operating is time-consuming, this is the main challenge faced in EMO-ELM. Thus, EMO-ELM is difficult to directly handle the big data. Fortunately, evolutionary algorithms are easy to implement in parallel. There is an issue that how to use EMO-ELM to do deep representation learning is worth studying in the future works.

## 5. Conclusions

This paper proposed an EMO-ELM approach for sparse feature learning of hyperspectral image. The main idea of EMO-ELM is that, firstly, using an EMO to optimize the hidden layer of ELM, then executing feature extraction according to the optimal hidden layer. To let EMO-ELM have the ability to learn sparse features, the hidden layer activations are constrained by an objective function. The other objective involved in EMO-ELM is the cross-validation RMSE, which ensures the learned features are accurate. Unlike ELM-AE which linearly transforms original features, EMO-ELM uses a nonlinear way to do this. This procedure is similar to neural network autoencoder, such AE, but EMO-ELM optimizes the hidden layer only. So that EMO-ELM can be updated using EMO. Experiments are carried out on two real hyperspectral image data sets, and the performance of EMO-ELM is extensively explored including the convergence, the sparsity, and the classification ability. According to the experimental results, we can draw the following conclusions:The experimental results demonstrate that EMO-ELMs is more suitable to extract sparse features from the hyperspectral image, and EMO plays a more significant role in dealing with the nonlinear data of hyperspectral image.The proposed EMO-ELM significantly improves the performance of the original ELM. These experimental results demonstrate that the optimized hidden layer of ELM is effective for HSI feature learning.EMO-ELM generally outperforms ELM-AE, SELM-AE, AE, and SAE in terms of sparsity and classification ability because of two optimized objectives.The knee-based solution selection strategy can accurately focus on the knee area of the PF curve. But, the RMSE-based solution selection strategy is more applicable in our experiments.

## Figures and Tables

**Figure 1 sensors-20-01262-f001:**
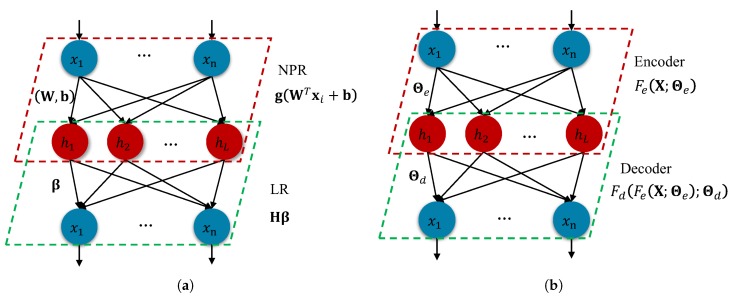
(**a**) ELM-AE includes NPR and LR. Using $\beta^{T}$ as the transformation matrix to transform features. (**b**) AE consists of an encoder (red rhomboid box) and a decoder (green rhomboid box). The outputs of encoder represent the learned features.

**Figure 2 sensors-20-01262-f002:**
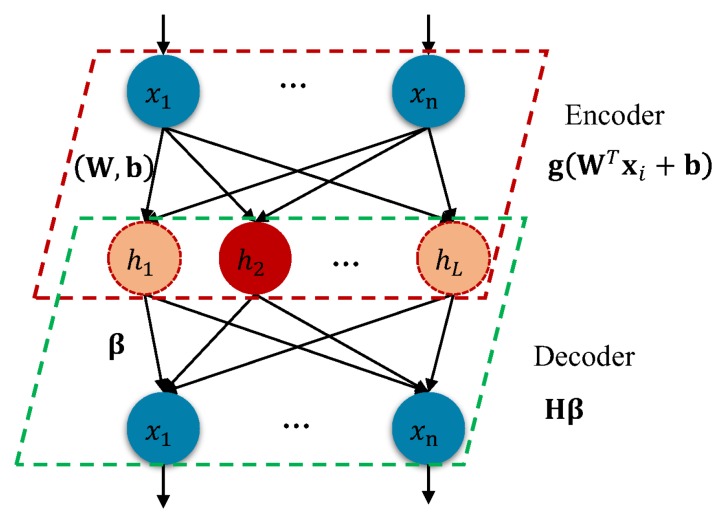
Structure of EMO-ELM which consists of an encoder (red rhomboid box) and decoder (green rhomboid box). Where the red neurons in hidden layer denote they are activated while the orange neurons represent they are limited.

**Figure 3 sensors-20-01262-f003:**
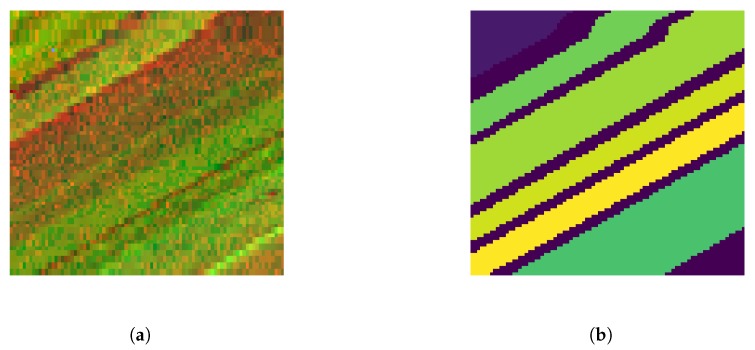
Pseudo-color image (**a**) and ground truth (**b**) of Salinas-A data set.

**Figure 4 sensors-20-01262-f004:**
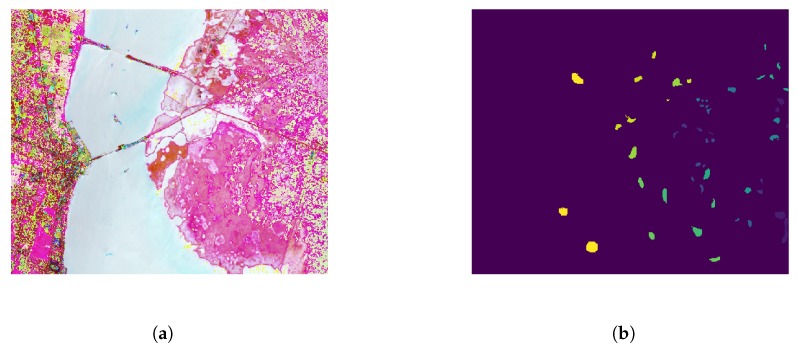
Pseudo-color image (**a**) and ground truth (**b**) of KSC data set.

**Figure 5 sensors-20-01262-f005:**
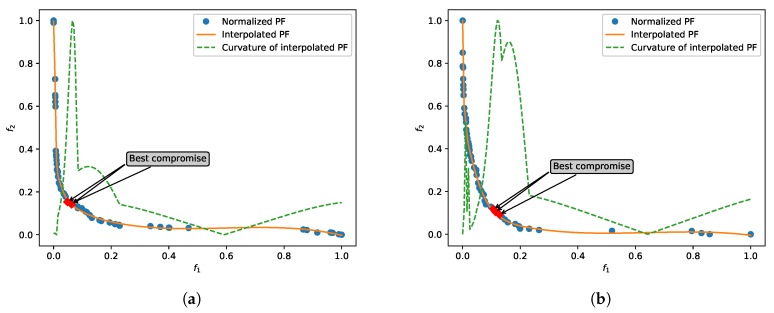
Normalized Pareto front and solution selection of (**a**) SalinasA and (**b**) KSC data sets. The curvatures are normalized for plotting it in a same coordinate. The best compromise is denoted as the top three points of closing to the maximum curvature.

**Figure 6 sensors-20-01262-f006:**
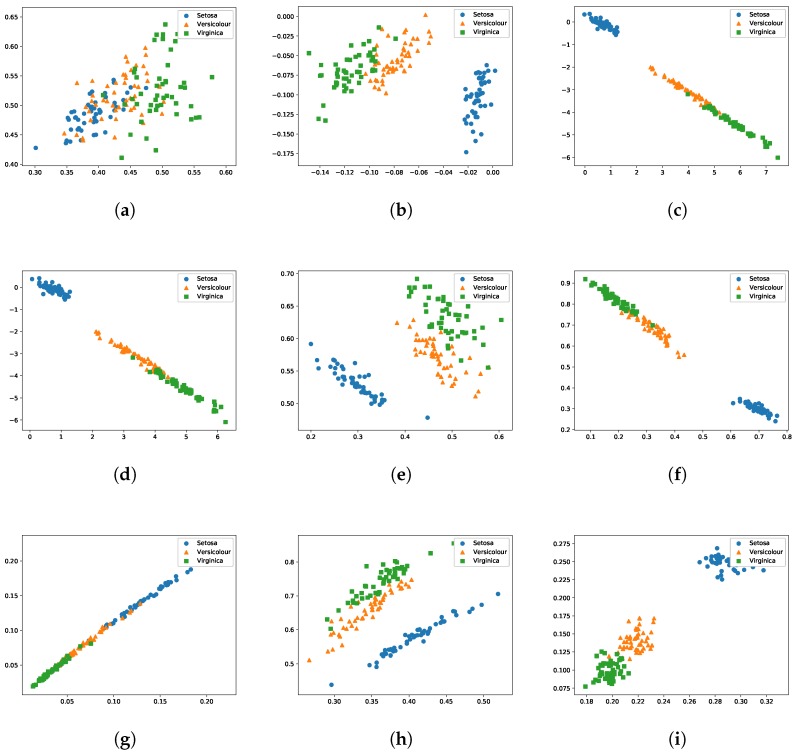
The 2-dimensional visualization of Iris dataset of (**a**) NRP, (**b**) SPCA, (**c**) ELM-AE, (**d**) SELM-AE, (**e**) AE, (**f**) SAE, (**g**) EMO-ELM($f_{1}$), (**h**) EMO-ELM($f_{2}$) and (**i**) EMO-ELM(best).

**Figure 7 sensors-20-01262-f007:**
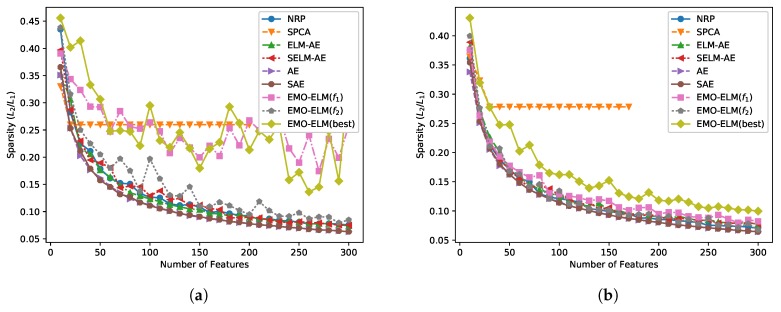
The sparsity of different algorithms of (**a**) SalinasA and (**b**) KSC data set.

**Figure 8 sensors-20-01262-f008:**
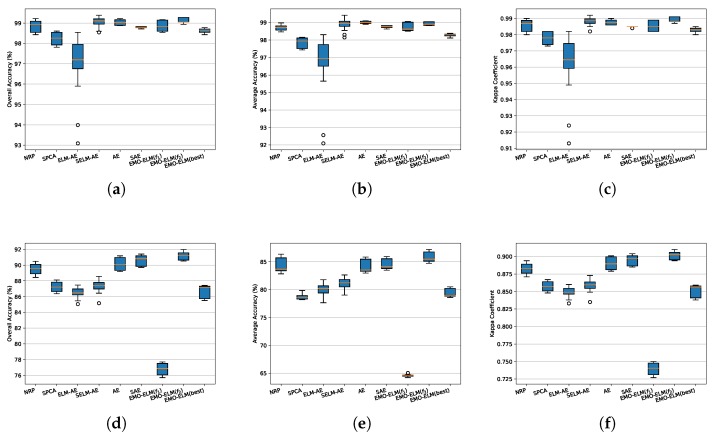
Box plot of SalinasA and KSC data sets. (**a**–**c**) denotes the box plot of the SalinasA data set in terms of OA, AA, and Kappa, respectively; (**d**–**f**) represents the box plot of the KSC data set with respect to OA, AA, and Kappa, respectively. The edges of boxes are the 25th and 75th percentiles and the middle lines indicate the median line. Whiskers extend to the maximum and minimum points. Abnormal outliers are shown as “∘”s.

**Table 1 sensors-20-01262-t001:** Groundtruth classes of the Salinas-A scene and their respective samples number.

#	Class	Number of Samples
1	Brocoli_green_weeds_1	391
2	Corn_senesced_green_weeds	1343
3	Lettuce_romaine_4wk	616
4	Lettuce_romaine_5wk	1525
5	Lettuce_romaine_6wk	674
6	Lettuce_romaine_7wk	799

**Table 2 sensors-20-01262-t002:** Groundtruth classes of the KSC scene and their respective samples number.

#	Class	Number of Samples
1	Scrub	761
2	Willow swamp	243
3	CP/Oak	256
4	Slash pine	252
5	Oak/Broadleaf	161
6	Hardwood	229
7	swamp	105
8	Graminoid marsh	431
9	Spartina marsh	520
10	Cattail marsh	404
11	Salt marsh	419
12	Mud flats	503
13	Water	927

**Table 3 sensors-20-01262-t003:** The performance comparison of SalinasA for NRP, SPCA, ELM-AE, SELM-AE, AE, SAE and EMO-ELM. (MEAN ± STD) L=10,iter=5000,ρ=0.05,NP=50.

Class	Algorithm								
	NRP	SPCA	ELM-AE	SELM-AE	AE	SAE	EMO-ELM($f_{1}$)	EMO-ELM($f_{2}$)	**EMO-ELM(best)**
1	**99.49 ± 0.72**	**99.49 ± 0.72**	99.16 ± 0.96	99.16 ± 0.94	**99.49 ± 0.72**	**99.49 ± 0.72**	**99.49 ± 0.72**	**99.49 ± 0.72**	**99.49 ± 0.72**
2	97.80 ± 0.44	98.73 ± 0.41	95.95 ± 2.28	98.81 ± 0.41	97.83 ± 0.58	97.57 ± 0.16	98.18 ± 0.64	**99.09 ± 0.43**	98.73 ± 0.40
3	96.12 ± 1.37	92.03 ± 1.96	86.54 ± 6.22	96.36 ± 1.59	**97.50 ± 0.80**	96.64 ± 0.74	96.14 ± 0.43	96.43 ± 0.83	92.94 ± 0.64
4	99.93 ± 0.09	99.8 ± 0.16	99.13 ± 1.91	99.72 ± 0.26	**100.00 ± 0.00**	99.87 ± 0.09	99.99 ± 0.04	**100.00 ± 0.00**	**100.00 ± 0.00**
5	**100.00 ± 0.00**	99.87 ± 0.2	99.44 ± 0.57	99.54 ± 0.57	99.70 ± 0.42	99.70 ± 0.42	99.99 ± 0.08	99.70 ± 0.42	99.81 ± 0.22
6	99.37 ± 0.47	97.12 ± 1.2	98.84 ± 0.44	99.20 ± 0.42	**99.50 ± 0.35**	99.25 ± 0.31	98.62 ± 0.34	98.89 ± 0.30	98.75 ± 0.71
**AA**	98.79 ± 0.23	97.84 ± 0.27	96.51 ± 1.29	98.80 ± 0.30	**99.00 ± 0.05**	98.75 ± 0.07	98.73 ± 0.22	98.93 ± 0.08	98.29 ± 0.07
**OA**	98.85 ± 0.16	98.22 ± 0.28	96.88 ± 1.28	98.96 ± 0.23	99.02 ± 0.14	98.78 ± 0.04	98.85 ± 0.23	**99.12 ± 0.12**	98.62 ± 0.09
**Kappa**	0.986 ± 0.002	0.978 ± 0.004	0.961 ± 0.016	0.987 ± 0.003	0.988 ± 0.002	0.985 ± 0.000	0.986 ± 0.003	**0.989 ± 0.002**	0.983 ± 0.001

**Table 4 sensors-20-01262-t004:** The performance comparison of KSC for NRP, SPCA, ELM-AE, SELM-AE, AE, SAE and EMO-ELM. (MEAN ± STD) L=10,iter=5000,ρ=0.05,NP=50.

Class	Algorithm								
	NRP	SPCA	ELM-AE	SELM-AE	AE	SAE	EMO-ELM($f_{1}$)	EMO-ELM($f_{2}$)	EMO-ELM(best)
1	97.76 ± 1.32	97.66 ± 0.66	96.57 ± 1.03	96.36 ± 1.10	97.83 ± 0.27	97.84 ± 0.49	95.11 ± 0.27	**98.09 ± 0.50**	96.19 ± 0.48
2	89.59 ± 3.81	92.10 ± 2.46	83.50 ± 3.63	85.72 ± 3.69	86.87 ± 2.60	86.71 ± 1.93	**94.40 ± 2.08**	89.09 ± 2.87	90.78 ± 0.83
3	88.32 ± 1.41	**93.73 ± 3.38**	90.12 ± 3.14	89.80 ± 3.20	92.13 ± 3.82	90.02 ± 4.44	92.97 ± 0.04	88.07 ± 3.59	91.68 ± 1.66
4	63.53 ± 1.42	31.15 ± 1.79	57.46 ± 5.57	56.19 ± 4.27	47.46 ± 2.06	51.43 ± 2.12	2.02 ± 0.55	**67.74 ± 3.26**	26.71 ± 5.96
5	55.58 ± 6.59	51.84 ± 4.34	43.50 ± 4.72	43.99 ± 3.73	58.15 ± 2.13	57.89 ± 2.81	16.92 ± 2.83	**58.68 ± 5.91**	54.23 ± 3.55
6	**54.09 ± 5.91**	29.48 ± 2.36	46.64 ± 7.24	48.78 ± 6.88	52.04 ± 5.98	47.96 ± 2.63	0.09 ± 0.33	50.41 ± 3.86	35.42 ± 3.37
7	**90.48 ± 5.39**	61.62 ± 10.32	82.29 ± 7.75	80.19 ± 9.67	86.10 ± 5.46	86.67 ± 8.32	41.05 ± 11.58	87.81 ± 7.63	77.81 ± 12.64
8	84.32 ± 4.96	81.24 ± 5.96	82.53 ± 6.43	83.67 ± 6.66	88.82 ± 6.13	**89.89 ± 4.29**	70.79 ± 3.40	88.59 ± 4.08	84.92 ± 6.83
9	97.06 ± 0.27	92.48 ± 0.99	96.62 ± 2.24	97.50 ± 1.69	97.56 ± 1.10	97.50 ± 1.32	79.76 ± 4.57	**98.81 ± 1.27**	97.94 ± 0.91
10	92.73 ± 1.28	**97.85 ± 1.60**	91.24 ± 2.88	90.55 ± 2.58	93.77 ± 2.38	96.49 ± 2.30	77.37 ± 1.41	94.66 ± 1.85	90.65 ± 1.87
11	98.59 ± 0.98	97.66 ± 0.36	94.41 ± 1.69	95.49 ± 1.81	98.83 ± 0.34	**99.16 ± 0.27**	83.60 ± 3.62	98.73 ± 0.36	93.89 ± 0.45
12	84.97 ± 1.53	95.63 ± 0.79	78.85 ± 2.98	81.63 ± 2.68	94.69 ± 0.96	96.28 ± 0.90	85.39 ± 1.05	**94.77 ± 1.73**	90.96 ± 1.74
13	99.81 ± 0.16	**100.00 ± 0.00**	98.34 ± 0.73	97.93 ± 0.59	**100.00 ± 0.00**	**100.00 ± 0.00**	99.40 ± 0.27	99.61 ± 0.19	99.32 ± 0.19
**AA**	84.37 ± 1.17	78.65 ± 0.38	80.16 ± 0.92	80.60 ± 1.11	84.17 ± 1.01	84.45 ± 0.85	64.53 ± 0.23	**85.77 ± 0.89**	79.27 ± 0.73
**OA**	89.51 ± 0.59	87.21 ± 0.60	86.49 ± 0.65	86.96 ± 0.72	90.20 ± 0.70	90.59 ± 0.56	76.77 ± 0.67	**91.21 ± 0.47**	86.70 ± 0.76
**Kappa**	0.883 ± 0.007	0.857 ± 0.007	0.849 ± 0.007	0.855 ± 0.008	0.891 ± 0.008	0.895 ± 0.006	0.739 ± 0.008	**0.902 ± 0.005**	0.851 ± 0.008

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
