# Peer review of "Sparse Feature Learning of Hyperspectral Imagery via Multiobjective-Based Extreme Learning Machine"

_sensors, 2020, doi:10.3390/s20051262_

Round 1
Reviewer 1 Report
The paper tackles the problem of learning sparse feature for Hyperspectral Imagery. The authors of this paper proposed a sparse unsupervised feature learning based on an extreme learning machine (ELM) and autoencoder (AE). The contribution consists of adapting an existing ELM-AE model by using Evolutionary Multiobjective Optimization(EMO) that allows improving the performance of ELM-AE by using non-linearity and reducing the number of parameters as compared to an autoencoder. The topic is highly relevant and the problem addressed is interesting. Good Points:
-The paper is well-motivated, and the problem is well defined. -The objective of the proposed approach and the obtained results are consistent -The use of the Evolutionary Multiobjective Optimization to optimize the hidden layer of ELM is interesting However, I have some comments to be addressed:
-A major concern is that the presentation of the key ideas of the research is deficient. In other words, the paper is hard to read and the passage from an idea to another is not very smooth. The writing need to be revised to improve the language quality, clarity, and readability.
-Examples of language-related errors: *L56-57:in combination of the advantages of AE and ELM, is proposed for unsupervised feature learning in hyperspectral imagery. *L69: a novel spares unsupervised feature learning approach *L80: ELM-AE adn SELM-AE *L116: and β is the optimizid solution. *L121: Xare represented as: The teta symbol can be better represented (ex: Eq 7) *etc -It would be easier if the authors detailed the AE limitations, then the ELM-AE limitations, then how their proposed EMO-ELM addresses these limitations in the introduction. The authors did this at the end of the introduction section in a brief way, but as mentioned earlier, the transition between the key ideas is weak, making it difficult and unpleasant to read the paper. -The quality of figure 8 is poor
Author Response
Please see the attachment named "reply-1".

Reviewer 2 Report
Please see the attached file

Author Response
Please see the attachment named "reply-2.pdf".

Round 2
Reviewer 1 Report
The authors answers my comments.
Reviewer 2 Report
Please see attached file
